# Exploration adhesion properties of *Liquorilactobacillus* and *Lentilactobacillus* isolated from two different sources of tepache kefir grains

**Julián Fernando Oviedo-León[1], Maribel Cornejo-Mazón[2], Rosario Ortiz-Hernández[3], Nayeli Torres-Ramírez[3], Humberto Hernández-Sánchez[1] \*, Diana C. Castro-Rodríguez[4] \***

1 Departamento de Ingeniería Bioquímica, Escuela Nacional de Ciencias Biológicas, Instituto Politécnico Nacional, Unidad Profesional Adolfo López Mateos, Mexico City, Mexico, 2 Departamento Biofísica, Escuela Nacional de Ciencias Biológicas, Instituto Politécnico Nacional, Carpio y Plan de Ayala, Mexico City, Mexico, 3 Departamento de Biología Celular, Facultad de Ciencias, Universidad Nacional Autónoma de Mexico (UNAM), Mexico City, Mexico, 4 Investigadores CONAHCyT, Departamento de Biología de la Reproducción, Instituto Nacional de Ciencias Médicas y Nutrición Salvador Zubirán, Mexico City, Mexico

\* diana.castror@incmnsz.mx (DCC-R); hhernan1955@yahoo.com (HH-S)

**Data Availability Statement:** All sequencing information can be found within GenBank at the

## Abstract

Due to the distinctive characteristics of probiotics, it is essential to pinpoint strains originating from diverse sources that prove efficacious in addressing a range of pathologies linked to dysfunction of the intestinal barrier. Nine strains of lactic acid bacteria were isolated from two different sources of tepache kefir grains (KAS2, KAS3, KAS4, KAS7, KAL4, KBS2, KBS3, KBL1 and KBL3), and were categorized to the genus *Lacticaseibacillus*, *Liquorilacto-bacillus*, and *Lentilactobacillus* by 16S rRNA gene. Kinetic behaviors of these strains were evaluated in MRS medium, and their probiotic potential was performed: resistance to low pH, tolerance to pepsin, pancreatin, bile salts, antibiotic resistance, hemolytic activity, and adhesion ability. KAS7 strain presented a higher growth rate (0.50 h⁻¹) compared with KAS2 strain, who presented a lower growth rate (0.29 h⁻¹). KBS2 strain was the only strain that survived the *in vitro* stomach simulation conditions (29.3%). Strain KBL1 demonstrated significantly higher viability (90.6%) in the *in vitro* intestine simulation conditions. Strain KAS2 demonstrated strong hydrophilic character with chloroform (85.6%) and xylol (57.6%) and a higher percentage of mucin adhesion (87.1%). However, strains KBS2 (84.8%) and KBL3 (89.5%) showed the highest autoaggregation values. In terms of adhesion to the intestinal epithelium in rats, strains KAS2, KAS3 and KAS4 showed values above 80%. The growth of the strains KAS2, KAS3, KAS4, KBS2, and KBL3 was inhibited by cefuroxime, cefotaxime, tetracycline, ampicillin, erythromycin, and cephalothin. Strains KBS2 (41.9% and 33.5%) and KBL3 (42.5% and 32.8%) had the highest co-aggregation values with S. aureus and E. coli. The results obtained in this study indicate that lactic acid bacteria isolated from tepache can be considered as candidates for potentially probiotic bacteria, laying the foundations to evaluate their probiotic functionality *in vivo* and thus to be used in the formulation of functional foods.

following accession numbers: OR077320 through OR077328 (https://www.ncbi.nlm.nih.gov/nuccore/?term=OR077320:OR077328[accn]).

**Funding:** The research that led to these findings received financial support from CONAHCYT (Consejo Nacional de Humanidades, Ciencias y Tecnologías, grant number 771244) and the Instituto Politecnico Nacional (grant number SIP20200277) in Mexico. The funders had no role in study design, data collection and analysis, decision to publish, or preparation of the manuscript.

**Competing interests:** The authors have declared that no competing interests exist.

# 1. Introduction

Stress, poor diet, and an unhealthy lifestyle are factors that cause serious gastrointestinal disorders, such as irritable bowel syndrome, gastroesophageal reflux and peptic ulcers, which affect a high percentage of the population. Microbiota plays an important role in human health, in such a way that its imbalance leads to dysbiosis and with it the appearance and progression of different pathologies, such as, the irritable bowel disease, antibiotic associated diarrhea, and the necrotizing enterocolitis. Probiotics interact closely with the host native microbiota and, therefore, share mechanisms of action that confer benefits to humans [1, 2].

The taxonomic restructuring of lactic acid bacteria within the *Lactobacillus* genus resulted in the creation of 23 new genera. These genera are distinguished by their varied metabolic properties influenced by the source of isolation. Notable among the newly established genera are *Lacticaseibacillus*, *Liquorilactobacillus*, and *Lentilactobacillus* [3]. Bacteria of the *Lacticaseibacillus* genus make up the largest percentage of commercial probiotics, and are characterized by their important role in the maintenance of the intestinal microbiota and in the stimulation of the immune system [4]. *Liquorilactobacillus*, a lactobacillus derived from liquid environments such as plant sap, water, and alcoholic beverages, has been identified as displaying tolerance to acid and bile concentrations mimicking human gastrointestinal conditions. This highlights its potential as a probiotic [5]. Bacteria of the *Lentilactobacillus* genus, known for their heterofermentative nature, have been employed to thwart aerobic spoilage in silage by converting lactic acid into acetic acid and 1,2-propanediol [6, 7].

Many *in vitro* properties, such as adhesion, resistance to gastrointestinal conditions, antimicrobial activity, among others, are routinely investigated to determine if a selected strain would be suitable as a probiotic [8]. In recent years, there has been an increasing interest in isolating new probiotics that have a great impact on human health. New strains with probiotic benefits have been isolated from various sources [9, 10], however, plant-derived fermented products have been a suitable source for the isolation of new probiotics [11–13].

In Mexico, there are about 200 fermented foods and beverages, which are strongly related to Mexican culture and traditions [14]. Among these traditional beverages with historical importance, is tepache. This is a fermented, pleasant drink that is consumed in Mexico. Tepache, which has been produced with corn and water since pre-Hispanic times, is thought to have the largest national traditional fermented beverage consumption [15, 16]. At the moment, tepache is made from agricultural wastes like pineapple peel or bagasse, and by adding tibicos, obtaining a low-alcoholic beverage. Tibicos are gelatinous masses made up of bacteria and yeasts that develop in sugary liquids kept at rest [17]. The origin of tibicos is not known for certain, but depending on the source, the composition of the bacteria and yeasts present in tibicos may vary [18, 19]. Water kefir, or sugary kefir or tepache of tibicos, contains 80–89% sucrose, 10% reducing sugars and about 0.4% protein [18]. In this study, tibicos, also called water kefir grains, were used as a source of isolation of new probiotics. They are made up of consortia of microorganisms embedded in a polysaccharide matrix, mostly lactic acid bacteria whose functional characteristics and safety profile make them potential probiotics of great interest in research.

# 2. Materials and methods

## 2.1. Isolation of LAB strains from tepache kefir grains

Tibicos or tepache kefir grains were obtained from two batches from an artisan tepache producer at a traditional market in Mexico City and State of Mexico, respectively. The kefir grains were gently separated from the drink using a strainer, recovering both the tibicos (solid phase)

and fermented product (liquid phase). Kefir grains were washed with sterile water; resuspended in 0.85% saline solution, proportion 1:3, and vigorously stirred for 10 minutes by vortex until disintegrated. On the other hand, fermented product (liquid phase) was centrifuged at 8000 X g for 15 min at 4˚C, recovering the pellet, which was resuspended in 10 mL of 0.85% saline solution. Each saline sample was seeded in MRS agar (Dibico, Mexico City) and incubated at 37˚C for 48 h. Subsequently, each morphologically distinct colony was seeded separately in MRS broth (Dibico, Mexico City) at 37˚C for 48 h. The above procedure was performed three times until the purity of the colonies was guaranteed. A first characterization of the isolated bacteria was performed by Gram staining, cell morphology and catalase reaction. All Gram-positive, rod-shaped and catalase-negative bacteria were stored in 20% glycerol at -70˚C for subsequent analysis of their probiotic properties. Activation of each bacterium was performed in MRS broth at 37˚C for 24 h [20].

## 2.2. Pathogen strains

To evaluate the antimicrobial activity of bacteria isolated from tepache kefir grains, the following pathogenic bacteria were used: *Staphylococcus aureus* subsp. *aureus* (ATCC 25923), *Listeria monocytogenes* strain Li 2 (ATCC 19115), *Escherichia coli* CDC EDL 933 (ATCC 43895), and *Salmonella enterica* subsp. *enterica* (ATCC 14028). All pathogenic strains were activated in Müeller-Hinton broth (Difco, Detroit, USA) at 37˚C under aerobic conditions.

## 2.3. Biochemical characterization

The isolates were identified by biochemical analysis using API 50 CHL kit assay (BioRad). The reading and interpretation of results was carried out 48 h after incubation at 37˚C under anoxic conditions and the ABIS online program was considered to determine the percentage of similarity (ABIS online—Bacterial identification (tgw1916.net)).

## 2.4. Bacterial identification by 16S rRNA sequences analysis

Total genomic DNA for each isolated strain was extracted according to QIAamp DNA Mini Kit Cat# 51304, the quality and concentration of DNA were assessed in NanoDrop 2000C, Spectrophotometer, Thermoscientific, Germany. The variable regions 3–4 of the 16S rRNA gene were amplified using specific forward (5″ TCGTCGGCAGCGTCAGATGTGTATAAGAG ACAGCCTACGGGNGGCWGCAG 3″) and reverse primers (5″ GTCTCGTGGGCTCGGAGAT GTGTATAAGAGACAGGACTACHVGGGTATCTAATCC 3″) containing the Illumina adapter overhang nucleotide sequences. The 16S V3-V4 amplicons were quantified using TapeSation (Agilent, Santa Clara, CA, USA) using Ampure XP bits to purify them. The amplicons were roughly 550 bp in size. The Nextera XT V2 Kit was then used to perform an index PCR in order to affix dual indices. The fluorometer Qubit 3.0 (Invitrogen, Waltham, MA, USA) was used to quantify the concentration of double-stranded DNA, and the amplicon size was roughly 630 bp. Equimolar concentrations of the final amplicon library were combined. The Illumina NovaSeq platform (Illumina, San Diego, CA, USA) was used for the sequencing to produce paired-end reads that were 250 bases long in each direction. The quality of the reads was checked with fastQC v0.12.1, the read mapping was constructed with bowtie2 version 2.5.1. The consensus sequence of each isolate was obtained with Ugene v46.0, using the reference sequence NR_025880.1 *Lacticaseibacillus paracasei* strain R094 16S ribosomal RNA, partial sequence. Sequences from isolated bacteria were compared with BLAST (http://blast.ncbi. nlm.nih.gov/Blast.cgi), and the phylogenetic tree was built with Mega v11.0.13.

## 2.5. Determination of fermentation characteristic

Homofermentative-heterofermentative differential agar (HHD) was used to evaluate acid production. Bacteria isolated from tepache kefir grains were incubated for 3 days at 37°C in this selective medium [21].

## 2.6. Growth kinetics

The strains were activated in MRS broth incubated at 37°C for 24 h. After, the strains were cultivated in MRS broth for 8 h at 37°C with pH 6.5. The number of bacteria was measured by optical density and plate count, at the time of inoculation (T0) and after each hour of incubation for 8 hours (T8). Moreover, the pH of the culture media was measured at the same time points. The assay was performed in triplicate. Specific growth rate (μ) was determined according to Maier et al., 2015 [22], for every strain cultivated in MRS broth.

## 2.7. Preparation of transmission electron microscopy (TEM) samples

Bacteria pellets were fixed in 2.5% glutaraldehyde in phosphate buffer (pH 7, 1X) for 2 h at room temperature. Next, the samples were rinsed with phosphate buffer and postfixed with 1% OsO4 for 1 h. The samples were dehydrated in graded series of ethanol from 30% to 100%. After dehydration, samples were infiltrated and embedded with Durcupan ACM resin (Sigma-Aldrich, St. Louis, MO). Ultrathin sections of 60 nm were cut using the ultramicrotome Leica Ultracut UCT, mounted on copper grids covered with formvar and contrasted with 4% uranyl acetate and 0.35% lead citrate. Sections were observed under a Jeol 1010 electron microscope at 80 kV. Digital images were captured with a Hamamatsu camera (Hamamatsu Photonics K. K., Japan). TEM images were analyzed using ImageJ software [23] to measure length, diameter and thickness of the peptidoglycan cell wall. The average length, diameter and thickness of the peptidoglycan cell wall were measured for 6 bacteria, in triplicate.

## 2.8. Viability under gastrointestinal conditions

The isolated strains from tepache kefir grains were activated in MRS broth for 24 h, and then $1x10^9$ CFU/mL of each strain were added to a flask with 50 mL of MRS broth at pH 2 with 6N HCl; pepsin (3 g/L); NaCl (2 g/L); $KH_2PO_4$ (0.6 g/L); $CaCl_2$ (0.11 g/L); KCl (0.37 g/L) and incubated at 37°C for 2h (stomach conditions) [24]. The strains were also inoculated in MRS broth supplemented with bile salts (3 mg/mL); pancreatin (2 g/L); $NaH_2PO4$ (50.8 g/L); NaCl (8.5 g/L), pH 6.5 and incubated at 37°C for 4 h (intestinal conditions) [25]. Each simulation was performed three times. The percentage of log viability was calculated as [Log ($CFU_{final}$/ mL)/Log ($CFU_{initial}$/mL)] x 100. The initial value corresponds to the amount of bacteria before being incorporated into stomach or intestinal conditions and the final value corresponds to the amount of bacteria obtained after incubation in simulated stomach or intestinal conditions [11, 26].

## 2.9. Adhesion properties

**2.9.1. Mucin adhesion.** The methodology described by Tallon et al., 2007 [27], Valeriano et al., 2014 [28] and Lara-Hidalgo et al., 2019 [29] was followed with some modifications. Porcine gastric mucin type II (Sigma-Aldrich) was dissolved in PBS (pH 7.2), obtaining a final concentration of 1 mg/mL. Then, 100 μL of this solution were added to a polystyrene microplate and incubated at 4°C overnight to achieve immobilization. Subsequently, each well was washed twice with 200 μL of PBS (pH 7.2). The wells were then saturated with 2% bovine serum albumin (20 mg/mL) for 4 hours at 4°C. After another wash with 200 μL of PBS. A 1

mL aliquot of each bacterial suspension was centrifuged at 3500 rpm for 5 min and the pellet formed was washed twice with 1 mL of PBS (pH 7.2) and resuspended by adjusting the concentration between $10^8$–$10^9$ CFU/mL. Subsequently, 100 μL of the bacterial suspension were taken and added to each well and incubated for 1 h at 37°C. The wells were washed 5–10 times using 200 μL of PBS (pH 7.2). Finally, 200 μL of 0.5% v/v Triton X-100 was added to each well to release the attached bacteria, incubated at room temperature with gentle agitation (300 rpm) for 2 h. For bacterial counts, an aliquot of 100 μL of the suspension was taken from each well and serial dilutions were performed for further analysis. All the runs were executed in triplicate. The log percentage of adhesion was determined according to the following formula: adhesion assay: M.A % = [Log ($CFU_{final}$/mL)/Log ($CFU_{initial}$/mL)] x 100.

**2.9.2. Auto-aggregation.** Auto-aggregation assay was performed using the nine strains under study according to Zulkhairi Amin et al., 2020 [30]. Each strain, after 24 h of growth in MRS broth, was centrifuged at 8000 X g and washed twice with PBS (pH 7.2). The initial concentration was adjusted to an optical density of 0.9 ± 0.05, equivalent to $1 \times 10^8$ CFU/mL. Each bacterial suspension (5mL) was shaken vigorously for 10 s and incubated at 37°C for 4 h. The percentage of auto-aggregation was calculated according to the following formulae: A % = $(1-A_t/A_0)$ x 100, where $A_t$ is the absorbance at 4 hours after incubation and $A_0$ is the initial absorbance before incubation. All the runs were executed in triplicate.

**2.9.3. Hydrophobicity.** The MATH (microbiological adhesion to hydrocarbons) method was used to evaluate the hydrophobicity of bacteria isolated from tepache kefir grains [31, 32]. The strains were cultured at 37°C for 24 h in MRS broth, then centrifuged at 13000 X g for 5 min, 20°C. The pellet obtained was washed twice with phosphate buffered saline, pH 7.2 (PBS). Three mL of bacterial suspension ($Abs_{560\ nm}$ = 0.23 ± 0.07) were added to 0.75 mL of chloroform or xylene. It was mixed gently in a vortex for 2 min and allowed to stand for 1 h at 37°C, subsequently the $Abs_{560\ nm}$ of the aqueous phase was evaluated. The percentage of hydrophobicity was determined as H % = [(Ai—Af)/Ai] x 100, where Ai corresponds to the initial absorbance (no solvent) and Af is the absorbance of the aqueous phase at the end of the test. Hydrophobicity was evaluated in triplicate.

**2.9.4. Adhesion to the intestinal epithelium.** The intestinal tissue adherence capacity was evaluated using large intestine of Wistar female rats of 350 g as model, conducted by Darmastuti et al., 2021 [33]; Brink et al., 2006 [34]; and Castro-Rodríguez et al., 2015 [11] with additional slight modifications. The large intestine was partitioned into nine approximately 1 cm$^2$ fragments for each isolate. Each fragment underwent ten washes with 0.9% w/w saline at 4°C. Subsequently, each fragment was incubated with the respective strain at a concentration of $1 \times 10^9$ CFU/mL in 1X PBS (pH 7.2) at 37°C for 3 hours, with gentle shaking at 150 rpm to promote adherence. Following incubation, each fragment underwent three washes with 1X PBS (pH 7.2). The tissue was scraped to count the viable bacteria in MRS agar. Each assay was independently conducted in triplicate. The adhesion percentage is expressed considering the difference between bacteria before and after contact with the tissue. % Ad = [Log (CFUadhered/mL)/Log (CFUinitial/mL)] * 100.

## 2.10. Antimicrobial activity

The bacterial strains were cultivated in MRS broth at 37°C for 24 h to evaluate their inhibitory action. Following incubation, the supernatants were kept and sterile filtrated using 0.42 μm syringe filters following centrifugation (8000 X g, 20 min, 20°C). The pH of the supernatants was raised to 6.5 with 1N NaOH to remove the inhibitory effects of lactic acid, and catalase (1 mg/mL) was added to completely remove the inhibition caused by the generation of $H_2O_2$. The agar diffusion technique was used to examine the supernatants. Pathogenic bacteria were

cultured overnight in Müeller-Hinton broth at 37°C, reaching a concentration of $1.2x10^8$ CFU/mL. Subsequently, they were spread onto Müeller-Hinton agar plates. Following this, wells with a diameter of 0.85 cm were carefully carved into each plate. Next, 25 μL of each supernatant were carefully introduced into the respective wells, and the plates were incubated for 24 h at 37°C. The evaluation of antimicrobial activity was conducted by measuring the diameters of the inhibition zones surrounding the wells. Each experiment was performed in triplicate [35].

## 2.11. Safety testing

**2.11.1. Antibiotic resistance.**   Bacteria obtained from tepache kefir grains were cultured in MRS broth at 37°C for 24 h. Subsequently, 100 μL portions of the bacterial suspensions, each containing $10^9$ CFU/mL, were introduced onto MRS agar plates at 30°C. Antimicrobial susceptibility test discs (multiplate Bio-RAD, Gram positive) were carefully placed on the surface of the agar [36]. The assessment of antibiotic resistance was performed by measuring the diameters of the inhibition zones. This experimental procedure was replicated three times, and the outcomes were categorized as sensitive (S) or resistant (R) based on the presence or absence of inhibition zones.

**2.11.2. Hemolytic activity.**   Bacteria isolated from tepache kefir grains were spread on Columbia agar plates, containing 5% (w/v) sheep blood, and incubated at 37°C for 48 h. Subsequently, each plate was examined to evaluate the type of hemolysis presented. β-hemolysis (clear zones around colonies), α-hemolysis (green colored zones around the colonies) or γ-hemolysis (no hemolysis, no clear zones around the colonies) [26].

**2.11.3. Coaggregation.**   The ability of strains isolated from tepache kefir grains to co-aggregate with two pathogen strains *Staphylococcus aureus* subsp. *aureus* (ATCC 25923) and *Escherichia coli* CDC EDL 933 (ATCC 43895) was evaluated according to Collado et al., 2008 [37]. Following a similar approach to the auto-aggregation assay, strains were incubated overnight at 37°C in MRS or BHI broth, as appropriate, and washed twice with PBS (pH7.2). Each bacterial group was resuspended to a concentration equivalent to an optical density ($OD_{600}$) of $0.9 \pm 0.05$ and mixed in equal proportions. Vigorous shaking was applied for 10 seconds and these strains were incubated at 37°C for 2 and 4 h, absorbance was measured at these two times. All experiments were carried out independently on three occasions. The percentage of coaggregation was determined using the following formula [38]: $[(A_{pat} + A_{probio})/2 -(A_{mix})/(A_{pat} + A_{probio})/2] x 100$, where $A_{pat}$ and $A_{probio}$ represent pathogen and probiotic strains of the separate bacterial suspensions in control tubes and $A_{mix}$ represents the absorbance of the mixed pathogen and probiotic strains at two times 2 and 4 h.

## 2.12. Statistical analysis

All statistical analyses were performed with SigmaPlot software. The data showed herein constitute the arithmetic means of values from three repetitions. Data are presented as mean ± standard error (SEM). Significant differences were established a significance level $p = 0.05$ by applying One-way ANOVA and Tukey tests [11].

## 3. Results

### 3.1. Morphological, biochemical, and physiological characterization

A total of nine lactic acid bacteria (LAB) were isolated from two batches from artisan tepache produced at a traditional market in Mexico City and State of Mexico. LAB isolates were identified through their morphological, biochemical and physiological features (Fig 1). Biochemical

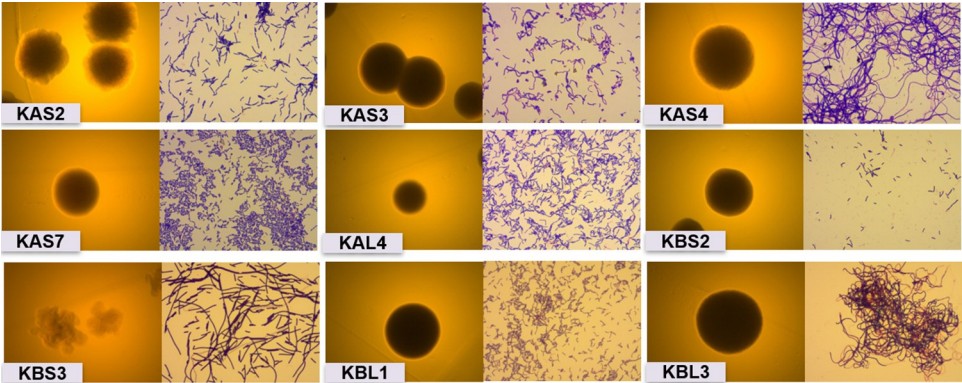

**Fig 1. Morphological identification by Gram stain and shape of the colony growing in MRS medium.**

properties evaluated by API 50 CHL kit assay, allowed the identification of these strains up to species level (S1 Table). All of them were Gram +, catalase -, and have a rod shape. Thus, the nine isolates were identified as members of the genus *Lactobacillus*, showing a similarity greater than 90% (S2 Table). Regarding growth and acid production in HHD agar medium, the KBS3 strain was the only heterofermentative, the other strains were homofermentative (S1 Fig).

## 3.2. Bacterial identification by 16S rRNA sequences analysis

Sequencing results on Novaseq 250 bp paired-end sequencing equipment can be observed in supporting information, as well as the bands of the PCR products. Based on 16S rRNA sequences analysis, the isolates were identified as strains of the lactic acid bacteria. Genus and species identified by each isolate, as well as the accession number in the GenBank database can be observed in Table 1. The neighbour joining algorithm is a method for constructing phylogenetic trees, based on the distance between species [39]. The phylogenetic tree of the nine isolates was grouped into four different genera, such as, *Liquorilactobacillus satsumensis*, *Lentilactobacillus hilgardii*, *Lacticaseibacillus casei* and *Lacticaseibacillus paracasei*, with *Leuconostoc mesenteroides* subsp. *mesentroides* as the outgroup (Fig 2), with differences between the sequences of each species of the same genus. The bootstrap value on the branches showed distances between the species, calculated statistically in 1000 replicates. The bootstrap value showed that the divergences or separation by these branches were correct or reliable. The quality of the sequences as well as the gels of the PCR products can be seen in the supporting information (S1 File).

**Table 1. Bacterial identification by 16S rRNA sequences analysis.**

| Isolate | Origin | DNA identification/accession number | Organism | Similarity (%) | Accession number |
|---------|--------|-------------------------------------|----------|----------------|------------------|
| KAS2 | Grain | *Lacticaseibacillus paracasei* strain VHProbi/ CP092498 | *Lacticaseibacillus paracasei* strain KAS2 | 98.59 | OR077320 |
| KAS3 | Grain | *Lactobacillus paracasei* strain GCUF-BNB-4/ KX388386.1 | *Lacticaseibacillus paracasei* strain KAS3 | 98.25 | OR077321 |
| KAS4 | Grain | *Lactobacillus paracasei* P7-1/ LC515428.1 | *Lacticaseibacillus paracasei* strain KAS4 | 98.06 | OR077322 |
| KAS7 | Grain | *Lactobacillus satsumensis*/ LC311746.1 | *Liquorilactobacillus satsumensis* strain KAS7 | 98.56 | OR077323 |
| KAL4 | Beverage | *Lactobacillus satsumensis*/ LC311746.1 | *Liquorilactobacillus satsumensis* strain KAL4 | 98.35 | OR077324 |
| KBS2 | Grain | *Lactobacillus casei* strain shebah-2/ MK829323.1 | *Lacticaseibacillus casei* strain KBS2 | 97.11 | OR077325 |
| KBS3 | Grain | *Lactobacillus hilgardii* strain UNQLh3.1/ KY561613.1 | *Lentilactobacillus hilgardii* strain KBS3 | 97.46 | OR077326 |
| KBL1 | Beverage | *Lactobacillus satsumensis*/ LC311746.1 | *Liquorilactobacillus satsumensis* strain KBL1 | 98.56 | OR077327 |
| KBL3 | Beverage | *Lactobacillus casei* strain BI-3.4/ KR135392.1 | *Lacticaseibacillus casei* strain KBL3 | 98.05 | OR077328 |

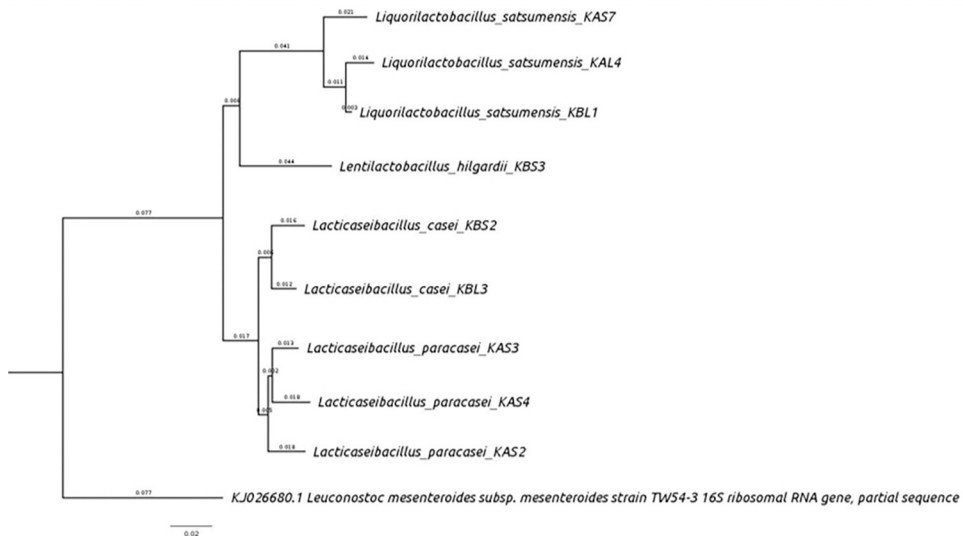

**Fig 2. Phylogenetic relationships of kefir isolates based on 16S rRNA gene sequences.** The tree was constructed by the neighbour-joining method. *Leuconostoc mesenteroides* was used as an outgroup.

## 3.3. Growth kinetics of the strains in MRS medium

Nine isolates from tepache kefir grains were studied for cell growth evaluation for incubation period of 8 h (Fig 3A). Following an 8-hour incubation period, the cell density derived from cultures of these strains in MRS ranged from 5.50 ± 0.02 log (CFU/mL) to 9.47 ± 0.06 log (CFU/mL). On the other hand, the strain KAS2 presented the lowest pH value (4.14 ± 0.02) compared with the strains KBL1 and KAS4, who presented higher pH values (5.60 ± 0.09 and 5.86 ± 0.03, respectively) (Fig 3A). In addition, KAS7 had the highest specific growth rate of 0.50 ± 0.04 h$^{-1}$ compared to KAS2, which had the lowest specific growth rate of 0.29 ± 0.03 h$^{-1}$ (Fig 3B).

## 3.4. Morphological characteristics by TEM

The morphological parameters of each of the isolated strains were measured, measuring the length and diameter of the bacteria, as well as the thickness of the peptidoglycan that makes up

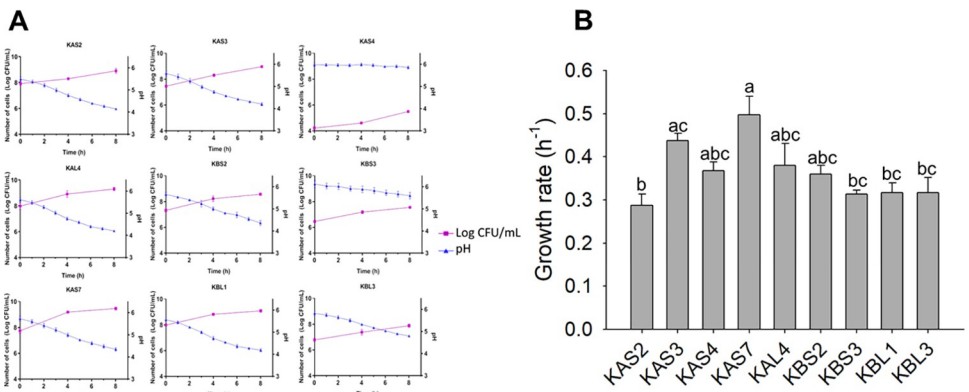

**Fig 3. Growth kinetics of the nine strains isolated of tepache kefir grains cultivated in MRS medium.** (A) number of cells (Log CFU/mL) and pH; (B) growth rate (h$^{-1}$). Each value is expressed as the mean ± SEM of three independent experiments (n = 3). Different letters are significantly different (P < 0.05), by applying ANOVA and Tukey test.

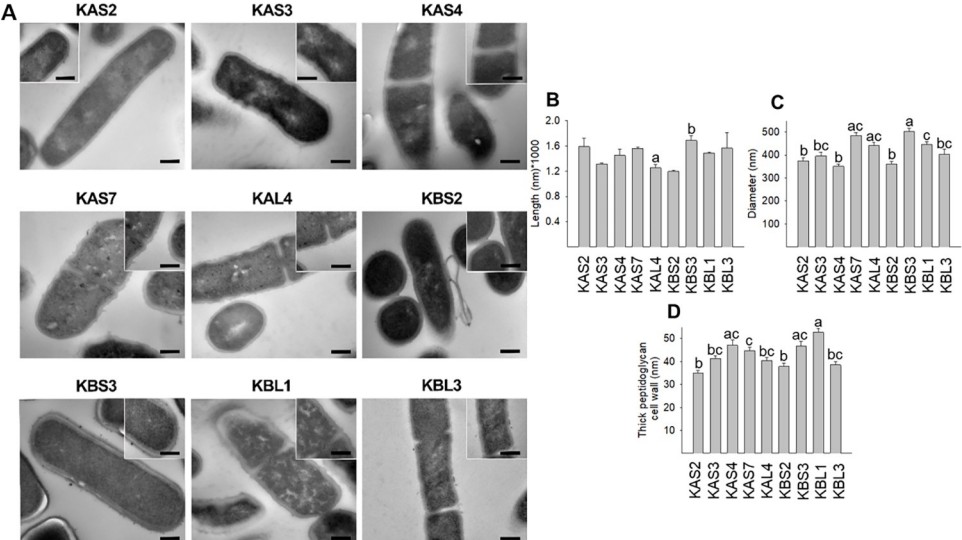

**Fig 4. Morphology and structure of bacterial cells under transmission electron microscopy (80000x-100000x).** (A) Morphology; (B) length; (C) diameter, and (D) thick peptidoglycan cell wall. Each value is expressed as the mean ± SEM of three independent experiments (n = 3). Different letters are significantly different (P < 0.05), by applying ANOVA and Tukey test. Scale bar: 200nm.

the cell wall of these Gram-positive bacteria (Fig 4A). Strain KBS3 had the highest length of 1685.18 nm (Fig 4B) and diameter of 501.68 nm (Fig 4C) compared to the other isolates. In terms of peptidoglycan size, strain KBL1 had the highest value (52.62 nm) compared to the other strains (Fig 4D).

## 3.5. Survival under conditions simulating the human gastrointestinal tract

The stomach conditions seriously affected the viability of the strains and only the strain KBS2 was able to survive these conditions (29.34 ± 0.30%) (Table 2). Under *in vitro* intestinal simulation conditions, the strains KAS2, KAS4, KAS7, and KAL4 did not survive. However, the strain KBL1 presented the highest percentage of viability (90.56 ± 0.41%) (Table 2).

**Table 2. Assessment of viability of the isolated strains after exposure to conditions simulating the human GI tract.**

| Strain | Stomach (%) | Intestine (%) |
|---|---|---|
| KAS2 | 0.00 | 0.00 |
| KAS3 | 0.00 | 68.33 ± 5.27[a] |
| KAS4 | 0.00 | 0.00 |
| KAS7 | 0.00 | 0.00 |
| KAL4 | 0.00 | 0.00 |
| KBS2 | 29.34 ± 0.30 | 69.35 ± 1.37[ab] |
| KBS3 | 0.00 | 77.35 ± 1.28[abc] |
| KBL1 | 0.00 | 90.56 ± 0.41[d] |
| KBL3 | 0.00 | 89.18 ± 1.11[cd] |

Each value is expressed as the mean ± SEM of three independent experiments (n = 3). Different letters are significantly different (P < 0.05) in same column, by applying ANOVA and Tukey test.

Table 3. Adhesion properties with mucin, autoaggregation, hydrophobicity, and intestinal epithelium of kefir isolates.

| Strain | Mucin adhesion (%) | Auto-aggregation at 4h (%) | Hydrophobicity (%) | | Intestinal epithelium (%) |
|---|---|---|---|---|---|
| | | | Chloroform (%) | Xylene (%) | |
| KAS2 | 87.12 ± 1.34[a] | 28.11 ± 2.49[bc] | 85.56 ± 2.61[a] | 57.65 ± 3.18[a] | 87.01 ± 2.00[a] |
| KAS3 | 65.05 ± 2.67[b] | 29.96 ± 0.77[bc] | 52.38 ± 2.32[b] | 27.35 ± 1.92[b] | 80.58 ± 1.81[ab] |
| KAS4 | 52.53 ± 4.96[bc] | 43.19 ± 9.23[b] | 66.96 ± 0.58[c] | 6.56 ± 3.13[c] | 83.55 ± 1.46[a] |
| KAS7 | 49.00 ± 0.92[c] | 15.25 ± 1.87[c] | 5.08 ± 0.26[d] | 0.19 ± 0.03[c] | 72.24 ± 1.52[b] |
| KAL4 | 51.51 ± 3.51[bc] | 18.22 ± 2.55[c] | 9.63 ± 0.84[d] | 0.21 ± 0.09[c] | 73.44 ± 3.00[b] |
| KBS2 | 67.33 ± 1.86[b] | 84.84 ± 1.75[a] | 83.32 ± 1.12[a] | 34.92 ± 1.81[b] | 72.57 ± 2.25[b] |
| KBS3 | 43.61 ± 3.35[c] | 27.07 ± 8.52[bc] | 32.98 ± 2.01[e] | 57.03 ± 2.75[a] | 77.41 ± 2.33[ab] |
| KBL1 | 53.37 ± 2.14[bc] | 14.35 ± 2.83[c] | 30.85 ± 4.33[e] | 2.17 ± 0.66[c] | 75.85 ± 2.50[b] |
| KBL3 | 46.62 ± 3.16[c] | 89.53 ± 3.86[a] | 77.24 ± 2.50[ac] | 4.55 ± 0.81[c] | 72.28 ± 0.69[b] |

Each value is expressed as the mean ± SEM of three independent experiments (n = 3). Different letters are significantly different (P < 0.05) in same column, by applying ANOVA and Tukey test.

## 3.6. Adhesion properties

**3.6.1. Mucin adhesion.** All strains showed greater than 40% adhesion, with KAS2 having the highest percentage of adhesion to gastric mucin (87.12%) compared to the other strains (Table 3). KAS3 and KBS2 presented values higher than 65%, being the second strains with the highest adhesion to mucin (Table 3). KBS3 showed the lowest percentage of adhesion (43.61%).

**3.6.2. Auto-aggregation.** The results of the auto-aggregation of the nine strains isolates are shown in Table 3. KBS2 and KBL3 showed the highest values 84.84 and 89.53% respectively, compared to the other strains which showed values below 50%.

**3.6.3. Hydrophobicity.** All strains have affinity for chloroform and xylene (Table 3). However, the strains KAS7 and KAL4 presented the lowest hydrophobicity values for both solvents. The strain KAS2 showed the highest affinity to chloroform and xylene (85.56 ± 2.61 and 57.65 ± 3.18%) compared to the other strains.

**3.6.4. Adhesion to the intestinal epithelium.** Each strain exhibited elevated adhesion values to the intestinal epithelium, as indicated in Table 3. Specifically, strains KAS2, KAS3, and KAS4 demonstrated the highest percentages, recording 87.01%, 80.58%, and 83.55%, respectively, compared to the other strains.

## 3.7. Antimicrobial activity

None of the isolates from the supernatant exhibited inhibition to the three enteropathogenic strains: *E. coli* ATCC43895, *S. enterica* ATCC 14028, and *L. monocytogenes* ATCC 19115, as well as the enterotoxigenic strain: *S. aureus* ATCC 25923.

## 3.8. Safety testing

**3.8.1. Antibiotic resistance.** The isolates displayed sensitivity to commonly used antibiotics in clinical applications but exhibited resistance to dicloxacillin. The growth of the strains KAS2, KAS3, KAS4, KBS2, and KBL3 was inhibited by cefuroxime, cefotaxime, tetracycline, ampicillin, erythromycin, and cephalothin (Table 4).

**3.8.2. Haemolytic activity.** All strains isolated from tepache kefir grains presented γ-hemolysis (no hemolysis), that is, they did not present any halo around the colonies. Thus, the safety of these strains is not of concern.

**Table 4. Antibiotic susceptibility in MRS agar for the isolates.**

| Antibiotic | KAS2 | KAS3 | KAS4 | KAS7 | KAL4 | KBS2 | KBS3 | KBL1 | KBL3 |
|---|---|---|---|---|---|---|---|---|---|
| Dicloxacilin 1 µg | R | R | R | R | R | R | R | R | R |
| Pefloxacin 5 µg | R | S | S | R | R | S | R | R | R |
| Cefuroxime 30 µg | S | S | S | R | R | S | S | R | S |
| Gentamicin 10 µg | R | R | R | R | R | R | S | R | R |
| Cefotaxime 30 µg | S | S | S | R | S | S | S | R | R |
| Trimethoprim 25 µg | R | R | R | S | S | R | S | S | R |
| Tetracyline 30 µg | S | S | S | S | S | S | R | S | S |
| Ampicillin 10 µg | S | S | S | S | S | S | S | S | S |
| Erythromycin 15 µg | S | S | S | S | S | S | S | S | S |
| Ceftazidime 30 µg | R | R | R | R | R | S | R | R | R |
| Cephalothin 30 µg | S | S | S | R | S | S | S | S | S |

S: sensitive, R: resistant

**3.8.3. Coaggregation.** The coaggregation values were similar in all strains between *S. aureus* after 2 h of incubation (Table 5), but after 4 h, the identified strains of the genus *Lactocaseibacillus* such as KAS2, KAS3, KAS4, KBS2 and KBL3 showed the highest co-aggregation between 25.49 and 42.50%. Regarding the co-aggregation between the isolates and *E. coli* strains, at 2 h KBS2 and KBL3 showed the highest value (26.28 and 26.68%, respectively). However, after 4 h KAS4, KBS2 and KBL3 showed the highest percentage of co-aggregation compared to the other isolates, ranging from 22.95 to 33.50% (Table 5).

# 4. Discussion

In this study, nine lactic acid strains were isolated from tibicos or tepache kefir grains, mostly identified as the genus *Lacticaseibacillus*, according to their biochemical properties, presenting similarities greater than 90% and their identification by 16S rRNA sequences analysis revealed that belonged to four species, including *Lacticaseibacillus paracasei*, *Liquorilactobacillus satsumensis*, *Lacticaseibacillus casei* and *Lentilactobacillus hilgardii*. All isolates strain with the exception of strain KBS3 were homofermentative. The strain KBS3 was identified as *Lentilactobacillus hilgardii*. The identification of the genus and species is very important for the

**Table 5. Percent coaggregation of strains isolates incubated in the presence of *Staphylococcus aureus* or *Escherichia coli*.**

| Strain | *Staphylococcus aureus* (%) | | *Escherichia coli* (%) | |
|---|---|---|---|---|
| | 2 h | 4 h | 2 h | 4 h |
| KAS2 | 11.43 ± 1.01 | 25.49 ± 0.35[b] | 8.03 ± 0.37[a] | 17.34 ± 0.88[a] |
| KAS3 | 16.29 ± 1.28 | 26.13 ± 2.12[b] | 10.17 ± 2.00[a] | 20.03 ± 1.41[a] |
| KAS4 | 24.76 ± 12.08 | 37.58 ± 9.47[b] | 11.42 ± 2.35[a] | 22.95 ± 0.33[b] |
| KAS7 | 10.97 ± 2.28 | 17.94 ± 1.22[a] | 10.19 ± 1.37[a] | 14.44 ± 1.02[a] |
| KAL4 | 7.49 ± 3.04 | 16.76 ± 0.99[a] | 6.77 ± 1.97[a] | 17.11 ± 2.64[a] |
| KBS2 | 29.02 ± 3.84 | 41.95 ± 2.49[b] | 26.28 ± 3.49[b] | 33.50 ± 0.94[b] |
| KBS3 | 16.31 ± 3.84 | 21.27 ± 6.62[ab] | 13.74 ± 6.10[ab] | 26.16 ± 5.99[ab] |
| KBL1 | 11.99 ± 1.20 | 17.72 ± 3.06[a] | 4.53 ± 3.23[a] | 15.32 ± 2.33[a] |
| KBL3 | 28.63 ± 2.92 | 42.50 ± 3.25[b] | 26.68 ± 2.41[b] | 32.76 ± 1.21[b] |

Each value is expressed as the mean ± SEM of three independent experiments (n = 3). Different letters are significantly different (P < 0.05) in same column, by applying ANOVA and Tukey test.

characterization of a probiotic strain, thus allowing precise and specific epidemiological studies of a particular bacterium [40]. Sequences of these isolates were deposited in NCBI database with sequence ID from OR077320 at OR077328.

The MRS commercial culture medium proves to be suitable for cultivating lactic acid bacteria from tepache kefir grains, as there is no observable adaptation phase from the time of inoculation. The pH values obtained during the kinetics analysis indicate that strains KAS2, KAS3, KAL4, KBS2, KAS7, and KBL1 exhibit promising potential for application in industrial processes, serving as essential starter microorganisms, either as primary cultures or as supplements [41]. Among these strains, KAS2, with its more substantial pH reduction compared to the others, holds particular promise for use in dairy product processing. This is because it can induce the destabilization of casein micelles, leading to coagulation through the solubilization of calcium phosphate [42]. The growth rate of a microorganism can be influenced by various factors, including nutrient concentration, pH, and temperature of the medium. In contrast to the specific growth rate reported by other authors, which stands at 0.53 ($h^{-1}$) for *Lacticaseibacillus* [43], the results from the strains isolated from tepache kefir grains indicate a lower specific growth rate.

TEM revealed structural differences between water kefir isolates. Differences between the size and shape of the strains could be observed. All strains displayed well-defined cytoplasmic contents, equal distribution of nuclear material and an intact cell-wall. The KAS3, KAL4 and KBL1 varied in size and shape and had a ruffled, intact and dense cell membrane. Morphologically similar to Gram-positive lactic acid bacteria isolated from vegetal origins [44, 45]. In addition, the advantage of examining cell morphology and structure allows to know the strains isolated from tepache kefir grains and to clearly identify changes in the overall morphology of the strains when subjected to stress.

This study provides evidence that strains isolated from tepache kefir grains, such as *Lacticaseibacillus*, exhibit the ability to withstand the harsh conditions of the gastrointestinal tract. To qualify as a probiotic, a strain must demonstrate resilience to the acidic environment of the stomach and the presence of bile salts in the intestine. Previous research has indicated that bacteria belonging to the *Lacticaseibacillus* genus, sourced from various origins, exhibit enhanced survival under intestinal conditions [20]. Additionally, other studies have highlighted the capacity of lactic acid bacteria to endure low pH levels [46, 47]. However, it is crucial to acknowledge that *in vitro* tests involving pH and bile salts may not fully replicate all the conditions encountered within the human body [48–50]. Bile salts play a significant role in the intestinal defense mechanisms, and their effects vary depending on their concentrations [51]. Physiologically, human bile contains bile salts in the range of 0.3% to 0.5% [52, 53].

It is important to highlight that adherence to the intestinal mucosa is a fundamental criterion to identify a probiotic bacterium; since this property allows this type of probiotic strains to colonize the intestine and increase their beneficial function in the host [54, 55]. Hydrophobicity could potentially serve as the initial point of interaction between microorganisms and host cells [56, 57]. The strains KAS2, KBS2, and KBL3 showed a higher hydrophobicity towards chloroform and xylene than other strains reported by other authors, which have isolated bacteria from various media such as fermented vegetables and dairy products, among others, obtaining different values depending on the bacteria (23.0 to 73.0%), *Pediococcus pentosaceus* CFRR38 and CFRR35, and *Lacticaseibacillus rhamnosus* GG ATCC 53510 (44.8 to 59.0%), and *Leuconostoc paramesenteroides* (46.11%) [56, 58, 59]. It is important to note that the percentage of hydrophobicity can vary depending on the conditions of the solvents used and the strains evaluated, which leads to different results.

Among the abilities of probiotic bacteria to adhere to the intestinal mucosa is their ability to adhere to mucin. Mucin is the mucus produced by the goblet cells that make up the intestinal

epithelium. It is a viscous material containing glycosylated proteins and is the most prominent point of interaction between microbes and humans, making it a prime target for controlling probiotic adhesion [60, 61]. The isolates presented adhesion percentages higher than 40% to gastric mucin in their majority, highlighting that strain KAS2 presented the highest percentage (87.12%), which makes it an excellent candidate to interact with the mucosa of the gastrointestinal tract, providing benefits to the host, thus allowing the possibility that these strains with probiotic potential play an active role in improving the intestinal barrier on the surface of the mucosa [62].

The auto-aggregating ability of probiotic strains is an important factor, as they can form a barrier and exclude pathogenic strains from adhering to the gastrointestinal tract [63]. Several authors have reported that this property depends significantly on the incubation time of the strains [64, 65]. For strains of the genus *Lacticaseibacilus* with 24 h of incubation reported auto-aggregation ranging from 50 to 70% [65, 66]. Strains of the same genus *Lacticaseibacillus*, but with a time of 5 h, the values were from 24 to 41% [67], and with 2 h values close to 25% [68] and 59% [69]. According to Rahman et al., 2008 [70], auto-aggregation values can be divided into high (>70%), medium (20–70%) and low (<20%). In our study, the auto-aggregation values obtained depend on the type of genus, we observed with a time of 4 h, high, medium and low auto-aggregations. Of the nine strains isolated from tepache kefir grains, the two strains *Lacticaseibacillus casei* (KBS2 and KBL3) showed high auto-aggregation, four strains *Lacticaseibacillus paracasei* (KAS2, KAS3 and KAS4) and *Lentilactobacillus hilgardii* KBS3 showed medium auto-aggregation and the three strains *Liquorilactobacillus satsumensis* (KAS7, KAL4 and KBL1) low auto-aggregation. Importantly, in our study, strains with low auto-aggregation were observed to have low hydrophobicity. These results could indicate that depending on the type of strain, hydrophobicity is one of the determinants of auto-aggregation, as reported by Rahman et al., 2008 [70], who finds a positive correlation between auto-aggregation and hydrophobicity in strains of the genus *Bifidobacterium*.

Different approaches have been employed to evaluate the adherence of probiotics, with models ranging from adherence to the intestinal epithelium to human colon carcinoma cells (Caco-2) [71]. These models closely resemble the conditions in which a probiotic bacterium is encountered upon consumption, evaluating host-specific factors such as health status or age. This approach yields more comprehensive information and is likely the most realistic option, as it considers the adherence of the normal microbiota present in the gut mucosa during the assay. In this investigation, remarkable adherence to the intestinal epithelium of Wistar female rats was noted for strains KAS2, KAS3, and KAS4 from the *Lacticaseibacillus* genus, with values surpassing 80%. This performance exceeded that of probiotic bacteria isolated from *Agave samiana* mead in the *Leuconostoc* genus [11], as well as bacteria from the same *Lacticaseibacillus* genus isolated from various fermented foods [72]. The observed adhesion characteristic supports the exclusion of pathogens and serves as a protective mechanism for epithelial cells. This adhesive property is ascribed to bacterial surface (S) layers, as well as other factors associated with the bacterial cell surface, including carbohydrates, lipoteichoic acids, and LPXTG-like protein factors [73].

Specifically, every strain isolated from tepache kefir grains met the crucial criterion for probiotic selection—the capacity to adhere to the intestinal mucosa (Table 3). Consequently, the findings from all the assessments indicate an outstanding adhesion capability, which is essential for successful colonization of the intestinal mucosa.

In this study, the supernatant obtained from the isolated strains showed low inhibitory activity. The inhibitory activity might be attributed to the bacteria's capacity to reduce the pH of the medium, leading to the production of different organic acids, including lactic and acetic acid, which possess bactericidal or bacteriostatic properties [74, 75].

In this research, various categories of antibiotics were employed, including cell wall inhibitors (Penicillin, Dicloxacillin, and Ampicillin), protein synthesis inhibitors (Tetracycline, Gentamicin, Erythromycin, Cephalothin, Cefotaxime, Ceftazidime, Cefuroxime), and inhibitors of DNA and RNA synthesis (Pefloxacin and Trimethoprim) [76]. The uncontrolled consumption of antibiotics can cause a dysbiosis (imbalance) of the microbiota, generating gastrointestinal problems. Probiotics do not present any safety problem when the resistance to antibiotics they possess is due to intrinsic resistance mechanisms, since they are responsible for the resistance phenotype and, therefore, intestinal eubiosis (balance) is preserved [77]. There are several types of resistance mechanisms, such as intrinsic or innate, acquired and mutational, that the same probiotic strain can present [78]. Bacteria can generate enzymes that inactivate antibiotics and thus develop antimicrobial resistance mechanisms [79, 80], an example of this is the production of β-lactamases by Gram-positive bacteria [81]. Strains obtained from tepache kefir grains exhibit resistance to dicloxacillin, pefloxacin, trimethoprim, and ceftazidime. These strains hold potential for aiding in the restoration of the intestinal microbiota in patients undergoing treatment with these specific antibiotics [82].

None of the strains isolated from tepache kefir grains showed hemolytic activity on blood agar, this is an indispensable requirement for a bacterium to be considered probiotic, as it will not generate any safety issues [30]. The results obtained in this study are similar to those reported by other authors where strains of *Lacticaseibacillus paracasei* subsp. *paracasei*, *Lacticaseibacillus* spp. and *Lacticaseibacillus casei* isolated from dairy products presented γ-hemolysis, but for a few that presented α-hemolysis [83].

One type of cell-cell interaction, known as coaggregation, is characterized by the highly specific recognition and adhesion of different species of microorganisms to each other [37, 84]. Coaggregation ability has mainly been characterized in bacteria from the human oral cavity and urogenital tract, the mammalian gut and drinking water supply systems [84]. It has been reported that the ability of lactic acid bacteria to coaggregation with pathogens may constitute a host defense mechanism against infection [37]. Therefore, the ability to co-aggregate with pathogens is an important probiotic property, which promotes the colonization of beneficial microorganisms [85, 86]. In this study, it is observed that all isolated strains have the ability to co-aggregate with the pathogens evaluated, but the percentages depend on each strain (both pathogenic and probiotic) and also on the incubation time [37]. It was observed that the longer the time, the greater the aggregation. Overall, the best probiotic strains with the best ability to co-aggregate with pathogens were *Lacticaseibacillus casei* KBS2 and KBL3, with these two having the highest autoaggregation percentages. The results are therefore consistent with the hypothesis that co-aggregation capabilities are related to autoaggregation properties [37].

## 5. Conclusion

The nine strains isolated from tepache kefir grains display intriguing probiotic characteristics, notably enhanced pH and bile tolerance, as well as hydrophobicity to chloroform and xylene, which were identified in *in vitro* tests as desirable probiotic traits. Moreover, they exhibit outstanding capabilities for adhering to both gastric mucosa and self-aggregation, two essential attributes of probiotic bacteria. In terms of safety assessment, these strains demonstrate the ability to coaggregate with potential pathogens such as *Staphylococcus aureus* ATCC 25923 and *Escherichia coli* ATCC 43895. The findings obtained thus far strongly suggest that these strains hold significant probiotic potential, serving as a foundation for further evaluations of their probiotic functionality *in vivo* and their potential application in the development of functional foods.

## Supporting information

**S1 Fig. Growth and acidification on homofermentative—heterofermentative differential (HHD) agar medium.**
(TIF)

**S1 Table. Biochemical profile evaluated by API 50 CHL kit assay.**
(PDF)

**S2 Table. Bacterial identification by ABIS online software, based on API 50CHL test.**
(PDF)

**S1 File. Bacterial 16S rRNA gene sequencing data.**
(PDF)

**S1 Graphical abstract.**
(TIF)

## Acknowledgments

We thank the Instituto Politecnico Nacional, the Instituto Nacional de Ciencias Médicas y Nutrición Salvador Zubirán and the Consejo Nacional de Humanidades, Ciencias y Tecnologías, all in Mexico City, Mexico.

## Author Contributions

**Conceptualization:** Humberto Hernández-Sánchez, Diana C. Castro-Rodríguez.

**Formal analysis:** Diana C. Castro-Rodríguez.

**Funding acquisition:** Humberto Hernández-Sánchez, Diana C. Castro-Rodríguez.

**Investigation:** Humberto Hernández-Sánchez, Diana C. Castro-Rodríguez.

**Methodology:** Julián Fernando Oviedo-León, Maribel Cornejo-Mazón, Rosario Ortiz-Hernández, Nayeli Torres-Ramírez, Diana C. Castro-Rodríguez.

**Writing – original draft:** Diana C. Castro-Rodríguez.

**Writing – review & editing:** Maribel Cornejo-Mazón, Humberto Hernández-Sánchez, Diana C. Castro-Rodríguez.

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
