## [Decision Letter · Decision Letter 0]

13 Nov 2023

PONE-D-23-31386Exploration adhesion properties of Liquorilactobacillus and Lentilactobacillus isolated from two different sources of tepache kefir grainsPLOS ONE

Dear Dr. Castro-Rodríguez,

Thank you for submitting your manuscript to PLOS ONE. After careful consideration, we feel that it has merit but does not fully meet PLOS ONE’s publication criteria as it currently stands. Therefore, we invite you to submit a revised version of the manuscript that addresses the points raised during the review process.

 Please carefully revise and answer the suggestions by the reviewers

We look forward to receiving your revised manuscript.

Kind regards,

Guadalupe Virginia Nevárez-Moorillón, Ph.D.

Academic Editor

PLOS ONE

Journal Requirements:

"The research that led to these findings received financial support from CONAHCYT (grant number 771244) and the Instituto Politecnico Nacional (grant number SIP20200277) in Mexico."

3. Please expand the acronym “CONAHCYT” (as indicated in your financial disclosure) so that it states the name of your funders in full.

"The research that led to these findings received financial support from CONAHCYT (grant number 771244) and the Instituto Politecnico Nacional (grant number SIP20200277) in Mexico."

"The research that led to these findings received financial support from CONAHCYT (grant number 771244) and the Instituto Politecnico Nacional (grant number SIP20200277) in Mexico."

"The authors declare that they have no conflict of interest."

7. Please include your tables as part of your main manuscript and remove the individual files. Please note that supplementary tables (should remain/ be uploaded) as separate "supporting information" files".

Reviewers' comments:

Reviewer's Responses to Questions

**Comments to the Author**

1. Is the manuscript technically sound, and do the data support the conclusions?

Reviewer #1: Yes

Reviewer #2: Partly

2. Has the statistical analysis been performed appropriately and rigorously? 

Reviewer #1: Yes

Reviewer #2: Yes

3. Have the authors made all data underlying the findings in their manuscript fully available?

Reviewer #1: Yes

Reviewer #2: Yes

4. Is the manuscript presented in an intelligible fashion and written in standard English?

Reviewer #1: Yes

Reviewer #2: Yes

5. Review Comments to the Author

Reviewer #1: Title: Exploration adhesion properties of Liquorilactobacillus and Lentilactobacillus isolated from two different sources of tepache kefir grains

o Reduce the Plagiarism up to 15 %

Abstract:

o The results are not efficiently summarized in the abstract. Write 2 line of research scope at start.

Introduction:

Mention at least some of the latest references related to Lentilactobacillus

The introduction is over generalized. Improve it add relevant references.

Anti-bacterial activity of essential oils against multidrug resistant foodborne pathogens isolated from raw milk

Isolation and Characterization of a Cholesterol-Lowering Bacteria from Bubalus bubalis Raw Milk

Antagonistic, Anti-oxidant, Anti-inflammatory and Anti-diabetic Probiotic Potential of Lactobacillus agilis Isolated From the Rhizosphere of the Medicinal Plants

Recent Innovations in Non-dairy Prebiotics and Probiotics: Physiological Potential, Applications, and Characterization

Materials and Methods:

Italicize the names of microorganisms.

Add sample collection perform as supplementary data

Line 103, Staphylococcus aureus (ATCC 25923), write subspecies as well with each pathogen

Line 259; containing 5% (w/v) human blood, or sheep bold? Cross check

Line 318, MRS v

Discussion:

Please add some latest references and improve the discussion further.

o Probiotics, their action modality and the use of multi-omics in metamorphosis of commensal microbiota into target-based probiotics

o Biosurfactant Screening and Antibiotic Analysis of Bacillus salmalaya

o Enhancing Lipase Production of Bacillus salmalaya Strain 139SI Using Different Carbon Sources and Surfactants

Seems over generalized the discussion part, kindly improve

References

Double check the references according to the Journal style.

Reviewer #2: The authors have analyzed the adhesive properties of Liquorilactobacillus and Lentilactobacillus isolated

from two different sources of tepache kefir grains in vitro which is a very basic study. In vitro adhesion properties may or may not replicate in vivo and for the development of probiotics or beneficial microbes, it is important that authors study the adhesion properties in appropriate cell lines and also in vivo in appropriate animals studies.

This study have reported only very basic in vitro characterization of few strains and to add the credibility and to understand the true probiotic potential authors should do in vivo studies as well.

6. PLOS authors have the option to publish the peer review history of their article (what does this mean?). If published, this will include your full peer review and any attached files.

Reviewer #1: No

Reviewer #2: No

---

## [Author Response · Author response to Decision Letter 0]

14 Dec 2023

Dear Editor, 

We are submitting a revised version of the manuscript: “Exploration adhesion properties of Liquorilactobacillus and Lentilactobacillus isolated from two different sources of tepache kefir grains” (PONE-D-23-31386) to be considered for publication in the journal PLOS ONE.

We appreciate the reviewers’ comments and address them below. The comments have improved the paper. We address the comments from the reviewers individually in the following section.

Comments of the reviewer #1:

1. Reduce the Plagiarism up to 15 % 

R: We appreciate your comment. We have corrected the manuscript to avoid similarity. We have attached the report made by Turnitin at the end of this document with 4% of plagiarism.

2. Abstract:

The results are not efficiently summarized in the abstract. Write 2 line of research scope at start.

R: Thank you for your comment, we have improved the presentation of the abstract (Lines 2-26). 

3. Introduction:

Mention at least some of the latest references related to Lentilactobacillus.

R: Thank you for your comment. We have mentioned the genus Lentilactobacillus and Liquorilactobacillus in the introduction (Lines 41-45; 48-54).

The introduction is over generalized. Improve it add relevant references.

Anti-bacterial activity of essential oils against multidrug resistant foodborne pathogens isolated from raw milk. 

Isolation and Characterization of a Cholesterol-Lowering Bacteria from Bubalus bubalis Raw Milk. 

Antagonistic, Anti-oxidant, Anti-inflammatory and Anti-diabetic Probiotic Potential of Lactobacillus agilis Isolated From the Rhizosphere of the Medicinal Plants

Recent Innovations in Non-dairy Prebiotics and Probiotics: Physiological Potential, Applications, and Characterization

R: Thank you very much for your suggestions. We have added them to improve the introduction.

4. Materials and Methods:

Italicize the names of microorganisms. Add sample collection perform as supplementary data.

R: Thank you for your comments. We have revised the names of the microorganisms. Regarding the data gathered for sample collection, it's important to note that the process is a unique, artisanal method and is not associated with any established collection or registered trademark.

Line 103, Staphylococcus aureus (ATCC 25923), write subspecies as well with each pathogen.

R: We have added the requested information (Lines 102-107).

Line 259; containing 5% (w/v) human blood, or sheep bold? Cross check.

R: Thank you for your comment. An apology for the error. We have corrected it in the manuscript: “containing 5% sheep blood” (Line 277).

Line 318, MRS v

R: We have rephrased the idea: “Following an 8-hour incubation period, the cell density derived from cultures of these strains in MRS ranged from 5.50 ± 0.02 log (CFU/mL) to 9.47 ± 0.06 log (CFU/mL)” (Lines 350-352).

5. Discussion:

Please add some latest references and improve the discussion further.

Probiotics, their action modality and the use of multi-omics in metamorphosis of commensal microbiota into target-based probiotics.

Biosurfactant Screening and Antibiotic Analysis of Bacillus salmalaya.

Enhancing Lipase Production of Bacillus salmalaya Strain 139SI Using Different Carbon Sources and Surfactants.

R: Thank you very much for your suggestions. We have added improved the discussion.

Seems over generalized the discussion part, kindly improve 

R: Thank you for your feedback. We have improved the discussion.

6. References:

Double check the references according to the Journal style.

R: Thank you for your comment, we checked the references according to the style of the journal.

Comments of the reviewer #2:

The authors have analyzed the adhesive properties of Liquorilactobacillus and Lentilactobacillus isolated from two different sources of tepache kefir grains in vitro which is a very basic study. In vitro adhesion properties may or may not replicate in vivo and for the development of probiotics or beneficial microbes, it is important that authors study the adhesion properties in appropriate cell lines and also in vivo in appropriate animals studies. This study have reported only very basic in vitro characterization of few strains and to add the credibility and to understand the true probiotic potential authors should do in vivo studies as well.

R: Thank you for your suggestion, which we have considered and evaluated adhesion to the intestinal epithelium in the colon of Wistar female rats with 350g as model. The methodology was added in the materials and methods section (Lines 232-245). The results were added in table 3 and in the results section (Lines 414-418). The results were discussed in the discussion section (Lines 565-587).

---

## [Editor Report · Decision Letter 1]

16 Jan 2024

Exploration adhesion properties of Liquorilactobacillus and Lentilactobacillus isolated from two different sources of tepache kefir grains

PONE-D-23-31386R1

Dear Dr. Castro-Rodríguez,

We’re pleased to inform you that your manuscript has been judged scientifically suitable for publication and will be formally accepted for publication once it meets all outstanding technical requirements.

Kind regards,

Guadalupe Virginia Nevárez-Moorillón, Ph.D.

Academic Editor

PLOS ONE
---

## [Editor Report · Acceptance letter]

30 Jan 2024

PONE-D-23-31386R1 

PLOS ONE

Dear Dr. Castro-Rodríguez, 

I'm pleased to inform you that your manuscript has been deemed suitable for publication in PLOS ONE. Congratulations! Your manuscript is now being handed over to our production team.

Kind regards, 

on behalf of

Dr. Guadalupe Virginia Nevárez-Moorillón 

Academic Editor

PLOS ONE